# A Dual-Signaling Electrochemical Aptasensor Based on an In-Plane Gold Nanoparticles–Black Phosphorus Heterostructure for the Sensitive Detection of Patulin

**DOI:** 10.3390/foods12040846

**Published:** 2023-02-16

**Authors:** Jinqiong Xu, Jianbo Liu, Wuwu Li, Yongsheng Wei, Qinglin Sheng, Yonghui Shang

**Affiliations:** 1College of Chemistry and Chemical Engineering, Xianyang Normal University, Xianyang 712000, China; 2College of Food Science and Technology, Northwest University, Xi’an 710069, China; 3Laboratory of Nutritional and Healthy Food-Individuation Manufacturing Engineering/Research Center of Food Safety Risk Assessment and Control, Xi’an 710069, China

**Keywords:** black phosphorous, in-plane, electrochemical aptasensor, dual-signaling, patulin

## Abstract

Patulin (PAT), a type of mycotoxin existing in foodstuffs, is harmful to food safety and human health. Thus, it is necessary to develop sensitive, selective and reliable analytical methods for PAT detection. In this study, a sensitive aptasensor based on a dual-signaling strategy was fabricated, in which a methylene-blue-labeled aptamer and ferrocene monocarboxylic acid in the electrolyte acted as a dual signal, for monitoring PAT. To improve the sensitivity of the aptasensor, an in-plane gold nanoparticles–black phosphorus heterostructure (AuNPs-BPNS) was synthesized for signal amplification. Due to the combination of AuNPs-BPNS nanocomposites and the dual-signaling strategy, the proposed aptasensor has a good analytical performance for PAT detection with the broad linear range of 0.1 nM–100.0 μM and the low detection limit of 0.043 nM. Moreover, the aptasensor was successfully employed for real sample detection, such as apple, pear and tomato. It is expected that BPNS-based nanomaterials hold great promise for developing novel aptasensors and may provide a sensing platform for food safety monitoring.

## 1. Introduction

Patulin (a polyketide lactone 4-hydroxy-4H-furo(3,2c) pyran-2(6H)-one, PAT) belongs to a class of toxic compounds known as mycotoxins and is a common contaminant in food and feed. It is a toxic metabolite of at least 60 different species of fungi, such as *Penicillium expansum* (*P. leucopus*), *P. patulum* (*P. urticae, P. griseofulvum*), *P. crustosum* and *A. clavatus* [1]. Therefore, humans are likely to be exposed to such toxic substances, which may cause various toxic effects, such as lung and liver bleeding, kidney damage, neurotoxicity, teratogenicity and possible carcinogenicity [2]. Accordingly, the levels of PAT in food are controlled by a set of regulations. Marín et al. [2] reported that the maximum level of patulin in pears and apples is 126 mg/kg in the Spanish market, which is twice as high as the European Union (EU) standard. The World Health Organization and the Food and Agriculture Organization recommend a maximum daily level of PAT of 0.4 μg/kg body weight/day [3]. In China, patulin is also a common contaminant in a variety of fruits and products. The rate of patulin detection was 76.9% in 401 fruit products from eight provinces in China [4]. Hence, different analytical techniques have been explored for highly sensitive and accurate detection of PAT, such as chromatography [5,6,7,8], colorimetry [9] and immunoassays [10,11]. However, these methods required heavy instrumentation and complex technology, especially concerning the instability of antibodies in immunoassays [12]. Electrochemical sensors are a powerful technique because of their fast response time, low cost and excellent flexibility.

At present, aptamers, a type of DNA or RNA, can recognize various target molecules (metal ions, small molecules, proteins, etc.) with high affinity and have drawn interest in sensor design [13,14]. Combining aptamers and electrochemical methods, electrochemical aptasensors possess an excellent performance. However, most electrochemical aptasensors have only one signal readout: either a “signal off” or “signal on”. The sensitivity of one-signal electrochemical sensors is inevitably limited, owing to signal fluctuation [15,16]. A dual-signaling strategy can solve these drawbacks. Numerous studies of electrochemical aptasensors with dual-signal channels have been reported to realize the high-performance detection of proteins and enzymes [17,18]. In dual-signaling electrochemical sensors, two electrochemical signals are dependent on targets, and their peak intensities show associated changes simultaneously. Therefore, dual-signaling electrochemical sensors exhibit a promising performance, with benefits such as a lower limit of detection and higher sensitivity.

To improve the sensitivity of electrochemical sensors, many nanomaterials have been applied for the amplification of signal. Among the various nanomaterials, two-dimensional (2D) nanomaterials have attracted widespread attention due to their unique physicochemical properties [19]. The first 2D nanomaterials appeared in 2004, when graphene was exfoliated from graphite by the mechanical cleavage technique [20,21]. Black phosphorus (BP), an allotrope of phosphorus with an orthorhombic lattice, is emerging as an important successor to 2D nanomaterials [22]. BP nanosheets (BPNS) can be exfoliated from bulk-layered crystal by breaking down its relatively weak van der Waals interactions, where each phosphorus atom forms covalent bonds with three neighboring atoms, forming a puckered orthorhombic structure [23,24]. The intriguing physical properties such as the high charge-carrier mobility, large on/off ratio, significant anisotropy and layer-dependent band gap have spurred tremendous interest in BP [25,26,27,28]. Owing to a direct, tunable band gap and no cytotoxicity, BPNS was used to explore photocatalyst and biomedical applications and more [29,30,31]. In order to improve the properties of BPNS and broaden their application, some nanocomposites based on BPNS have been synthesized by hybridization, doping or functionalization [32]. Sun et al. developed BPNS/graphene for sodium-ion batteries, which was self-assembled by intercalating BPNS between graphene layers via van der Waals interactions [33]. Hersam and co-workers reported that BPNS was functionalized by aryl diazonium through the formation of P–C covalent bonds and applied for field effect transistors [34]. Currently, the application of BPNS or BPNS-based nanocomposites in electrochemical sensors has not been adequately explored, which may be because of the instability of BP. Our group prepared gold nanoparticles–BPNS by electrostatic attraction for PAT electrochemical sensing [35]. However, this preparation method requires a more intensive experimental process and cannot enhance the stability of BP. The preparation of monodispersed BP nanosheets depends on long-time ultrasonic treatment or shear exfoliation, during which the BP layers are gradually cleaved and defects are generated on the edges and surface. These defects not only lead to the rapid degradation of the BP nanosheets, but also undermine the conductivity and electrochemical activity, thus causing problems in many applications—especially long-term catalysis. It is a good strategy to design the heterostructures to tailor the properties of 2D nanomaterials [36]. Due to the intrinsic advantages of the in-plane heterostructures, the in-plane charge carrier mobility of 2D materials is much higher than that between interlayers.

In this work, an in-plane gold nanoparticles–BPNS (AuNPs–BPNS) heterostructure was synthesized, in which the high chemical activity of the defect sites in BPNS can edge-selectively reduce HAuCl_4_, and the formed AuNPs not only occupy the defect sites but also provide effective immobilization sites for the aptamer. Then, the obtained AuNPs–BPNS nanocomposites were used to construct a simple dual-signaling electrochemical aptasensor for sensitive and accurate detection of PAT (Figure 1). The dual-signal was obtained from a methylene blue (MB)-labeled aptamer and ferrocene monocarboxylic acid (FMCA) in the electrolyte. The MB-labeled PAT aptamer self-assembled on the surface of pre-prepared AuNPs–BPNS/glassy carbon electrode (GCE) through the Au–S bond. Then, single-stranded DNA complementary to the aptamer formed a rigid double strand. In the presence of PAT, the aptamer recognized PAT and the complementary DNA was released. As a result of the conformational change in the aptamer, the MB tag came close to the surface of the electrode and the diffusion of FMCA was hindered due to the increased steric resistance. Thus, the peak current of MB increased and FMCA decreased, producing “signal on” and “signal off”. In this sensing system, because of the electronic property of BPNS and the sensitivity of the dual-signaling strategy, the proposed aptasensor exhibited an excellent performance for PAT detection. Moreover, the prepared aptasensor could also be applied in real sample detection. The developed sensing platform exhibited a great potential for the sensitive detection of other aptamer-specific binding contaminants of foodstuffs.

## 2. Materials and Methods

### 2.1. Chemicals and Reagents

The BP crystals were purchased from HWRK Chemical Co., Ltd. (Beijing, China) and stored in a glove box under dark. Chloroauric acid (HAuCl_4_) was purchased from Acros Organics (Shanghai, China). *N*-methyl-2-pyrrolidone (NMP) was purchased from Aldrich (Shanghai, China). PAT, tris(2-carboxyethyl)phosphine hydrochloride solution (TCEP) and 6-mercaptohexanol (MCH) were purchased from Sangon Biotech. Co., Ltd. (Shanghai, China). All other solvents were of analytical grade. All solutions were prepared with ultrapure water (>18 MΩcm) from a Millipore Milli-Q water purification system.

The PAT aptamer and DNA sequence employed in this research were purchased from TaKaRa (Dalian, China). The sequences were as follows:

PAT aptamer:

5′-MB-CCCGGCCCGCCAACCCGCATCATCTACACTGATATTTTACCTTCCC-SH-3′.

Complement DNA (cDNA): 5′-AAAGGGTTGGCGGGCC-3′.

### 2.2. Apparatus and Characterization

All of the electrochemical experiments were performed with an IGS1130 electrochemical workstation (Guangzhou Insens Sensor Technology Co. Ltd, Guangzhou, China). The three-electrode system consisted of a modified GCE (3 mm in diameter) as the working electrode, a platinum wire as the auxiliary electrode, and Ag/AgCl (saturated KCl) electrode as the reference electrode; this system was used in all electrochemical investigations. Transmission electron microscopy (TEM) images were acquired with a Hitachi H-7500 high-resolution transmission electron microscope (Tokyo, Japan) using an accelerating voltage of 200 kV. UV–Vis absorption spectra were acquired on a Cary 50 Scan UV–Vis spectrophotometer (Varian, Australia). X-ray photoelectron spectra (XPS) were obtained with an ESCALAB 250 Xi XPS system from Thermo Scientific (Waltham, MA, USA). Raman spectra were recorded using a Renishaw Micro-Raman spectroscopy system (London, England) with an excitation wavelength of 514 nm.

### 2.3. Preparation of BPNS

For preparation of BPNS, 5 mg of BP crystal was dispersed in 50 mL saturated sodium hydroxide solution of NMP under sonication at room temperature for 12 h. Then, the solution was centrifuged at 4000 rpm for 10 min and the supernatant was collected. Finally, BPNS were obtained by centrifuging the supernatant at 12,000 rpm for 15 min and dispersed in water as a brown solution.

### 2.4. Preparation of AuNPs–BPNS

AuNPs were modified on BPNS by in situ reduction. Briefly, 250 μL of 25 mM HAuCl_4_ was added to 1 mL of 1 mg/mL BPNS solution and stirred for 1 h in oxygen-free dark conditions. The products were separated by centrifugation and washed with deionized water. Finally, the obtained product was dispersed in water and stored at 4 °C.

### 2.5. Preparation of the Modified Electrodes

The GCE was polished to a mirror-like surface by 1.0 and 0.3 μm alumina slurries, respectively, and completely cleaned in ethanol and ultrapure water by ultrasonication. Then, 10 μL of a 1 mg/mL AuNPs–BPNS solution was dropped into GCE and dried in air at room temperature. Before the aptamer was fixed, the aptamer was added to phosphate-buffered saline solution (PBS; pH 7.0) containing 10 mM TCEP for treatment for 1 h to reduce the presence of disulfide bonds, and then the treated aptamer was dropped onto AuNPs–BPNS/GCE for 12 h. The electrode was then immersed in 1 mM MCH for 1 h to block the unbound sites. Finally, 10 μL of 1 μM cDNA was dropped onto the electrode surface for 1.5 h to obtain the final modified electrode cDNA/aptamer/AuNPs–BPNS/GCE.

### 2.6. Electrochemical Detection of PAT

For the detection of PAT, the modified electrode was immersed in 20 μL of a fixed concentration of PAT for 60 min under room temperature and then rinsed with the PBS to wash away the unbound PAT. The obtained electrode was then immersed in PBS (pH 7.4) containing 1 μM FMCA and investigated by cyclic voltammetry (CV) and cyclic voltammetry (SWV). The selective experiments for hydroxymethylfurfural (HMF), aflatoxin B1 (AFB1), aflatoxin M1 (AFM), ochratoxin A (OTA) and zearalenone (ZEN) were performed under the same conditions. All experiments were performed at room temperature and measured three times.

### 2.7. Pretreatment of Food Samples

In this experiment, fresh and infected apples, pears, and tomatoes acted as real food samples. The pretreatment procedures were performed according to previous work [37]. Apples, pears and tomatoes were smashed separately, then 2.5 g of the obtained sample mash was added into 10 mL H_2_O and extracted by ultrasonication for 1 h. The supernatant from each sample was transferred to a centrifuge tube and centrifuged at 6000 rpm for 5 min. Then, the extracts were collected. Finally, the supernatant and different concentrations of PAT were added to PBS.

## 3. Results

### 3.1. Characterization of the AuNPs–BPNS

The morphologies of the as-prepared BPNS and AuNPs–BPNS were characterized by TEM technologies. The TEM micrograph of BPNS is presented in Figure 1A. BPNS presents a smooth surface with some wrinkles, which is a 2D nanostructure. Figure 1B shows the TEM patterns of AuNPs–BPNS nanocomposites. AuNPs were successfully synthesized on the surface of BPNS without other reduction and the inset of Figure 1B shows an average diameter of AuNPs of about 21 nm. Elemental analyses of the AuNPs–BPNS nanocomposites entailed EDX mappings of P, Au and O, where P and O originated from BPNS and Au indicated the presence of AuNPs (Figure 1C). Notably, AuNPs were uniformly distributed on the entire BPNS surface. As shown in Figure 1D, the EDX analysis further revealed the existence of P, Au and O in the nanocomposites.

The effect of HAuCl_4_ concentration on the morphology of AuNPs–BPNS was also investigated. As shown in Figure 2A, when 10 mM HAuCl_4_ was added, only a small amount of AuNPs were generated on the surface of the BPNS. As the HAuCl_4_ concentration increased to 20 mM (Figure 2B), the amount of AuNPs on the BPNS surface increased. A satisfying amount of AuNPs–BPNS could be obtained once the concentration of HAuCl_4_ was increased to 25 mM (Figure 2C). It can be seen that AuNPs were uniformly deposited on the surface of BPNS without obvious aggregation, which may be because the defect of BPNS is exactly occupied by the AuNPs. However, when 30 mM HAuCl_4_ was added (Figure 2D), the AuNPs showed obvious aggregation. Therefore, 25 mM HAuCl_4_ was selected to prepare the AuNPs–BPNS nanocomposites.

UV–Vis spectroscopy was conducted to characterize the formation process of the AuNPs. Figure 3A shows the UV–Vis spectrum of BPNS with different amounts of added HAuCl_4_. As the amount of BPNS increased, the absorption peaks of the AuNPs increased at 536 nm, demonstrating that the formation of the AuNPs is related to the BPNS. Additionally, the UV–Vis spectrum of HAuCl_4_ added to BPNS with the increase in mixing time is shown in Figure 3B. The absorption peak intensity of the AuNPs increased with the increase in time, and the color of the solution changed to different shades of a red color. The successful synthesis of AuNPs may be due to the edge-selective reduction of BPNS. Moreover, AuNPs can occupy the defects in BPNS to enhance the stability of BPNS [38]. Raman spectroscopy could provide detailed vibrational and rotational patterns for sample identification. Raman spectra of BPNS and AuNPs–BPNS are provided in Figure 3C. Three peaks are displayed at 363 cm^−1^, 440 cm^−1^ and 466 cm^−1^, which were ascribed to one out-of-plane phonon mode A_g_^1^ and two in-plane phonon modes B_2g_ and A_g_^2^ for BP, respectively [39,40]. Upon AuNPs’ modification, the three peaks blue-shifted slightly because of vibration inhibition. Particularly, the B_2g_ and A_g_^2^ mode shift was larger than that of the A_g_^1^ mode, indicating that AuNPs–BPNS possess an in-plane heterostructure. Furthermore, the changes in oxidation states of BPNS from pristine BPNS to AuNPs–BPNS were probed by XPS (Appendix A). In Figure 3D, curves a and b display the P2p spectra before and after AuNP modification, respectively. The two unchanged peaks near 128.5 and 129.5 eV verify the original state of BP, which is consistent with standard BP. The peak at 133 eV is due to oxidized phosphorus (i.e., PO_x_) [41,42,43]. However, for AuNPs–BPNS (Figure 3Db), the PO_x_ peak was enhanced, demonstrating that some BP was oxidized in the formation of AuNPs–BPNS. The experimental data coincide with those reported in the literature [38].

### 3.2. Characterization of the Modified Electrode

EIS is an efficient method for exploring the interfacial properties of the modified electrodes [44,45]. Therefore, the modification process was characterized by EIS to reveal the electrodes’ surface features. EIS measurement was performed in 5.0 mM K_3_[Fe(CN)_6_]/K_4_[Fe(CN)_6_] (1:1) solution containing 0.1 M KCl. The equipment was set at 5 mV of alternative voltage amplitude and 200 mV of applied potential, and the voltage frequencies ranged from 0.01 Hz to 100 kHz. The electron transfer resistance (*R_et_*) was expressed by the diameter of a semicircle. As shown in Figure 4A, the bare GCE showed a small semicircle diameter (177 Ω, curve a), indicating a fast electron-transfer process. When BPNS was immobilized on the GCE, the semicircle diameter became bigger than that of bare GCE (704 Ω, curve b). As a result of the good electron transfer efficiency of the AuNPs, the AuNPs–BPNS/GCE exhibited a lower *R_et_* (566 Ω, curve c) than BPNS/GCE, suggesting that AuNPs could efficiently enhance the electron transfer efficiency. After the assembly of the aptamer, the value of *R_et_* increased to 1123 Ω (curve d) because the negatively charged aptamer and [Fe(CN)_6_]^3-/4-^ anions repelled each other. With the hybridization between the aptamer and cDNA, the semicircle diameter further increased (curve e), which suggests the successful recognition of the aptamer and cDNA. The results of EIS verified that the aptasensor was successfully prepared (Figure 1). Furthermore, XPS spectra for the C, N, Au and S energy regions are shown in Appendix A, which also verified the aptamer immobilization on AuNPs–BPNS.

The SWV measurements were carried out to investigate the electrochemical behavior of cDNA/aptamer/AuNPs–BPNS/GCE in different conditions. As shown in Figure 4B, only a low oxidation peak of MB at −0.25 V (curve a) can be observed in PBS without FMCA. After the addition of FMCA into PBS, a significant oxidation peak was produced at 0.3 V (curve b). However, after incubation with PAT, the MB response increased significantly and the FMCA response decreased (curve c). The results reveal that the proposed aptasensor could be used for PAT detection. PAGE analysis was performed to demonstrate the DNA combination processes step by step. As shown in Appendix A, Lanes 1 and 2 correspond to the aptamer and cDNA, respectively. Lane 3 represents the mixture of aptamer and cDNA; a new band was observed, demonstrating the hybridization among aptamer and cDNA. After PAT was added to this mixture, the band of hybridization weakened but the band of cDNA was obvious, which can be attributed to the initiation of hybridization and the release of cDNA. It implies that the aptamer and cDNA hybridized successfully and the aptamer could recognize PAT; thus, the cDNA was released.

### 3.3. Electrochemical PAT Assay

In order to obtain an excellent performance in sensing, some parameters were optimized: (a) the concentration of aptamer, (b) the hybridization time of aptamer and cDNA, (c) the incubation time of PAT. Respective data and figures are given in the electronic Supporting Material (Appendix A). The following experimental conditions were found to give the best results: (a) optimal aptamer concentration: 1 μM; (b) optimal hybridization time: 90 min; (c) optimal incubation time: 60 min.

The SWV was implemented to study the analytical performance of the proposed aptasensor. As shown in Figure 5A, a series of different concentrations of PAT from 0.1 nM to 100 μM were measured in PBS containing 10 μM FMCA. The peak current of MB increased and that of FMCA decreased with increasing PAT concentration. Figure 5B displays the linear relationship between the individual MB or FMCA current response and the logarithmic values of PAT concentration. For the MB signal, the limit of detection (LOD) was 0.011 ng/mL with R = 0.9987, and for the FMCA signal, 0.012 ng/mL with R = 0.9967. Furthermore, from Figure 5C, the linear relationship between the value of I_MB_/I_FMCA_ and the logarithm of the PAT concentration was acquired as follows: I_MB_/I_FMCA_ = 0.12 log C_PAT_ + 0.38 with R = 0.9941. A good linear correlation was obtained from 0.0154 ng/mL to 15.4 μg/mL and the LOD reached 0.066 ng/mL (S/N = 3). Compared with either an MB or FMCA signal sensor, the dual-signal strategy has a lower LOD.

Moreover, the analytical performance for the proposed dual-signaling aptasensor is comparable with other methods in previous reports as can be seen in Table 1. This may be because BPNS has an excellent electronic performance and two-dimensional sheet structure with a large surface area which provides more sites for the generation of AuNPs and prevents AuNPs aggregation; in addition, the dual-signal ratio strategy greatly reduces the impact of the external environment on the detection results and improves the sensitivity of the sensor. The results indicate that the proposed aptasensor exhibits a lower LOD and wider linear range.

The linearity in real spiked matrix samples was determined using pretreated fresh apple juice containing 10 μM FMCA and different PAT concentrations. The calibration curve (Figure 6) was obtained from the linear regression of a plot of peak area versus concentration. Calibration was carried out in the range of 0.154–15,400 ng/mL, being found to be linear in the entire studied range with R = 0.9956.

### 3.4. Reproducibility, Stability and Selectivity

The reproducibility of the proposed aptasensor was evaluated by using five cDNA/aptamer/AuNPs–BPNS/GCEs for the detection of 0.154 μg/mL PAT under the same conditions (Figure 7A). The relative standard deviation (RSD) was calculated to be 4.5%, revealing significant reproducibility. Moreover, the aptasensor was stored under refrigeration for ten days. As shown in Figure 7B, 91.1% of the initial response of the current was obtained after adding the same amount of PAT (0.154 μg/mL), demonstrating the satisfactory stability of the proposed aptasensor.

Furthermore, five kinds of contaminants in food samples containing hydroxymethylfurfural (HMF), aflatoxin B1 (AFB1), aflatoxin M1 (AFM), ochratoxin A (OTA) and zearalenone (ZEN) were used to verify the selectivity of the aptasensor. As shown in Figure 8, despite the fact that the concentrations of the interferences were as high as 10 μM, there were low values of I_MB_/I_FMCA_. However, an obvious response was obtained by the addition of PAT. The results indicated that the proposed dual-signaling aptasensor exhibits an excellent selectivity for PAT.

### 3.5. Real Sample Detection

To evaluate the practical application capability of the present system, the proposed aptasensor was employed to detect the PAT contents in fresh and infected apple, pear, and tomato. As shown in Table 2, no PAT was found in the fresh foods, while small amounts were found in the infected foods. Then, the standard addition method was also performed. The standard PAT solutions were added into the food samples; the recoveries and RSDs were 95.0–106.7% and 2.9–5.1%. Furthermore, there was no significant difference between the results obtained by the aptasensor and an HPLC method, as shown in Table 1. These results implied that the dual-signaling aptasensor is applicable for PAT detection in food samples.

## 4. Conclusions

In summary, BPNS have potential value in the field of food safety detection. Metal nanoparticles, especially AuNPs, as an efficient catalyst, promote electron transfer on the AuNPs–BPNS, overcoming the defects of single metal nanoparticles. In this paper, in-plane gold nanoparticles–BPNS (AuNPs–BPNS) was synthesized, in which the high chemical activity of the defect sites in BPNS can edge-selective reduced HAuCl_4_, and the formed AuNPs not only enhance the stability of BPNS but also provide effective immobilization sites of the aptamer. The obtained AuNPs–BPNS nanocomposites were used to construct a dual-signaling aptasensor for PAT detection. Under optimum conditions, the proposed aptasensor was linear over the range of 0.0154 ng/mL–15.4 μg/mL. The electrochemical sensing platform has been successfully applied to the determination of PAT in apple, pear, and tomato extracts with good accuracy and reliability. Food safety detection methods based on functional nanomaterials will usher in new development opportunities in the future.

## Data Availability

The data are available from the corresponding authors.

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
