# Peer review of "A Dual-Signaling Electrochemical Aptasensor Based on an In-Plane Gold Nanoparticles–Black Phosphorus Heterostructure for the Sensitive Detection of Patulin"

_foods, 2023, doi:10.3390/foods12040846_

Round 1

Reviewer 1 Report

The manuscript titled “Dual-Signaling Electrochemical Aptasensor Based on In-Plane Gold Nanoparticles-Black Phosphorus Heterostructure for Sensitive Detection of Patulin” was reviewed for consideration in Foods. The methodology is appropriate and topic is of interest.

Line 26-27: Please include which class belongs to PAT? And discuss the maximum permissible limits of that toxin in food.

Methodology

No quality parameters were included for detection of PAT. Like confirmation with HPLC etc.

Conclusion: The conclusion part must be revised and should be written again.

Reviewer 2 Report

The manuscript foods-2191417 entitled “Dual-signaling electrochemical aptasensor base on in-plane gold nanoparticles-black phosphorus heterostructure for sensitive detection of patulin” presents a good Introduction with adequate References, together with a clear organization of the Materials and Methods and the Results sections. However, I doubt its originality, since the authors published another very similar article in 2019: Microchimica Acta 186(4) 1-8. It is also an aptasensor based on AuNPs and BPNSs for the same application, the determination of PAT. The analytical characteristics obtained in this previous publication are similar to the new ones: LoD of 0.03 nM, linear range 0.1 nM-10 µM, recovery range 96.2-104 % with RSD less than 5%. I do not see a noticeable improvement that justifies another publication.

Reviewer 3 Report

This manuscript describes the development of an electrochemical sensor for monitoring of patulin. The developed sensor using gold nanoparticles-black phosphorus heterostructure (AuNPs-BPNS) was applied for signal amplification based on dual-signaling SWV measurements. The work is interesting and demonstrated desirable analytical performances for detection of patulin in food. The following issues could be addressed.

1.       Table 1: Authors should follow the rules concerning significant digits. The final number of significant digits should be given to the shortest number of significant digits in the numbers we are multiplying or dividing.

2.       Figure 3 caption (B) SWV measurents ==> “measurements “

3.       Repeatability of response: How is the repeatability of the electrode? No data are given. Can the sensor be reused?

4.       The author claimed that the sensor has good analytical performance such as wide linear range, low LOD and satisfactory reproducibility, stability and selectivity. So what is the limitation(s) of the proposed method?

Reviewer 4 Report

the introduction part should be reduced approx 30%. authors should be focused on the most important information about the patulin and materials for its extraction.

the concentration of contaminants and mycotoxins should be mentioned in standard units according to the legislation. It means no micro mols but microgram per liter.

line 261 - how many runs are possible to use the aptamer sensors for a standard solution and for real matrix samples?

what about the stability after 10 days? is the at same level of approx. 90%?

 line 141 - which previous work? please add te reference.

line 146 - what is the abbreviation PBS?

line 250 - the linearity should be tested in real spiked matrix samples to compare the matrix effects.

Concerning the interference study. Because hydroxymethylfurfural is very often presented in fruit juices after heat treatment, the interference experiment should be done with this analyte.

Round 2

Reviewer 2 Report

I do not see a noticeable improvement that justifies another publication.